# Indoor Positioning Algorithm Based on Reconstructed Observation Model and Particle Filter

**Li Ma [1,2], Ning Cao [1,*], Xiaoliang Feng [3], Jianping Zhang [4] and Jingjing Yan [2]**

[1]   School of Computer and Information, Hohai University, Nanjing 211106, China; mail_ml@haut.edu.cn
[2]   College of Electrical Engineering, Henan University of Technology, Zhengzhou 450052, China; yanjingjing2009@haut.edu.cn
[3]   School of Electrical Engineering, Shanghai Dianji University, Shanghai 201306, China; fengxl@sdju.edu.cn
[4]   School of Information Engineering, Zhengzhou Institute of Technology, Zhengzhou 450044, China; 20051010@zzut.edu.cn
[*]   Correspondence: 20020009@hhu.edu.cn; Tel.: +86-1862-371-8207

**Abstract:** In a complex indoor environment, wireless signals are affected by multiple factors such as reflection, scattering or diffuse reflection of electromagnetic waves from indoor walls and other objects, and the signal strength will fluctuate significantly. For the signal strength and the distance between the unknown nodes and the known nodes are a typical nonlinear estimation problem, and the unknown nodes cannot receive all Access Points (APs) signal strength data, this paper proposes a Particle Filter (PF) indoor position algorithm based on the Kernel Extreme Learning Machine (KELM) reconstruction observation model. Firstly, on the basis of establishing a fingerprint database of wireless signal strength and unknown node position, we use KELM to convert the fingerprint location problem into a machine learning problem and establish the mapping relationship between the location of the unknown node and the wireless signal strength, thereby refocusing construct an observation model of the indoor positioning system. Secondly, according to the measured values obtained by KELM, PF algorithm is adopted to obtain the predicted value of the unknown nodes. Thirdly, the predicted value is fused with the measured value obtained by KELM to locate the position of the unknown nodes. Moreover, a novel control strategy is proposed by introducing a reception factor to deal with the situation that unknown nodes in the system cannot receive all of the AP data, i.e., data loss occurs. This indoor positioning experimental results show that the accuracy of the method is significantly improved contrasted with commonly used PF, GP-PF and other positioning algorithms.

**Keywords:** indoor positioning; kernel extreme learning machine; particle filter; reconstructed observation model

## 1. Introduction

Indoor positioning has been extensively used and developed with the increasing demand for indoor position information in recent years. Many places, such as the location of workers in the mine, patients in hospitals, and pedestrians in shopping malls, require accurate location information of targets. Although the Global Positioning System (GPS) for positioning in outdoor environments is relatively mature [1–3], GPS signals are easily blocked by buildings in indoor surrounding and the signals are very weak. Therefore, indoor positioning has pay closer and closer attention. Nowadays, the demand for indoor positioning is increasing, and wireless positioning systems in different environments have been widely studied. Indoor positioning systems based on wireless signals have become a research hotspot in last several years [4–7]. Indoor positioning is broadly classified into two categories according to the positioning principle: infrastructure-less and infrastructure-based, both of which have achieved good research results.

The infrastructure-less approach does not require the support of existing infrastructure or networks, such as APs. For example, reference [8] proposed a smartphone-based indoor

localization method with an accelerometer and gyroscope, which can be used to find the distance traveled and heading estimation of pedestrians by accelerometer and gyroscope sensors respectively. Reference [9] proposed a problem of localization and navigation for blind people in indoor environments using processing and sensing techniques of smartphones rather than relying on external technologies. Reference [10] proposed a crowd sourced landmark indoor localization method based on accelerometer, magnetometer and gyroscope sensors of smartphones and adaptively optimized landmark algorithms are used correct the drift errors caused by sensor readings with the corresponding landmark database established by the experimental environment.

Infrastructure-based approaches require the use of existing infrastructure, such as Wi-Fi, and have achieved many successful research results. Reference [11] predicts the location coordinates of indoor robots based on RSS Kalman localization algorithm. Reference [12] investigated a weighted least squares based indoor localization system. Reference [13] proposed a particle filtering multi-target tracking algorithm based on RSS measurements for wireless sensor networks. The algorithm firstly uses approximate Least Squares (LS) for initial localization, and then completes the whole multi-target tracking by PF. Reference [14] studied the application of PF in indoor localization, and such PF method can meet the requirements of low-cost localization. To improve location accuracy of the PF indoor positioning technique, it is necessary to obtain an accurate transmission model of each Wi-Fi access point, which is more difficult.

The above references [11–14] focus on positioning methods by ranging based on RSS propagation models. The received RSS measurements are usually converted into distance measurements by the shaded logarithmic propagation model, and then the LS, Kalman, and PF methods are used for position estimation. From the shaded logarithmic propagation model, it can be seen that the signal strength and the distance between the unknown nodes and the known nodes have a logarithmic nonlinear relationship, which is a typical nonlinear estimation problem [15]. In addition, due to the complex indoor conditions, wireless signals are affected by diffuse reflection, scattering or reflection, bypassing, refraction, or transmission of electromagnetic waves from objects such as building walls in a complex indoor environment, which produces a certain impact on the propagation model. These methods are affected by the environment. Then it is difficult to accurately determine the target model of unknown nodes, and thus many errors still exist.

Compared to propagation model methods, which are highly influenced by the environment, location fingerprint-based indoor localization methods [16,17] can reduce the impact of signal shadow fading and multipath effects. Fingerprinting techniques have been used for both infrastructure-less and infrastructure-based, such as fingerprinting localization methods by the physical quantity of Received Signal Strength Indication (RSSI). Location fingerprint-based indoor localization methods have also made great research progress. References [18,19] developed a set of Android applications which are based on the RSSI of Wi-Fi to obtain the location by the Weighted K Nearest Neighbor (WKNN) algorithm. Reference [20] proposed a spatial-based WKNN indoor localization algorithm feature partitioning and previous location restrictions and divides the localization large area into multiple partitions based on their spatial features to solve the problem that one fingerprint library cannot achieve full coverage. Reference [21] proposed a Maximum Likelihood Particle Filter (MLPF) that can reduce the number of particles in indoor dynamic localization to produce highly accurate localization. Reference [22] discussed an indoor positioning algorithm based on fuzzy logic. It used the RSSI of the Bluetooth beacon and the geometric distance from the current beacon to the fingerprint point to calculate the Euclidean distance and then determined the position in the fuzzy logic framework. Reference [23] proposed a novel iBeacon beacon placement strategy, which can achieve 21.7% higher accuracy than the existing common iBeacon placement. In reference [24], a new Wi-Fi positioning method is proposed which is fusing RSSI derived fingerprints and multiple classifiers. Reference [25] put forward an Extreme Learning Machine (ELM) localization algorithm technique in view of RSSI. The Gaussian Process model enabled the

PF (GP-PF) algorithm for device-free localization algorithm is proposed in reference [26]. References [18–28] are about fingerprint localization methods. These methods are based on the case where an unknown node can receive all AP signal strengths.

For the problem of blocked wireless signals in complex indoor environments, which leads to the inability to receive all APs data, this paper proposes a control strategy by introducing a reception factor E, which can solve the problem of unknown nodes not being able to receive all APs data due to blocked wireless signals in real environments. Meanwhile, we adopt the KELM method to transform the fingerprint localization problem into a machine learning problem. On the basis of reconstructing the non-linear mapping relationship between the indoor positioning of the unknown nodes and the wireless network, we combine PF to make the localization system more robust to measurement noise.

We simulate the PF indoor localization method based on the KELM reconstructed observation model and actual Wi-Fi indoor localization experiments. Under the same conditions, we compare it with existing localization algorithms such as PF and GP-PF. The experimental results show that the KELM-PF algorithm proposed in this paper improves the positioning accuracy of unknown nodes and is more suitable for the positioning of unknown nodes in complex environments.

The four basic structures of this paper are described briefly as follows: The second section introduces KELM theory. The proposed reconstructed observation model indoor positioning method is described in detail in the third section. The indoor positioning experimental results and analysis are described in the fourth section. Finally, conclusions and prospects are illustrated in the fifth section.

## 2. Kernel Extreme Learning Machine

ELM is a machine learning algorithm based on Single Layer Feed Forward Neuron Network (SLFN) proposed by Professor Huang Guangbin [29,30]. There are three layers of architecture generally, including the input, hidden and output layers, to solve the problems of cumbersome parameter setting of the backpropagation algorithm and low learning efficiency. Huang Guangbin et al. [31,32] proposed KELM by solving the process between ELM and vector machine. The KELM algorithm is obtained by the kernelization of the ELM method, which is a SLFN with a kernel function. Compared with ELM, KELM is more robust and performs better in linearly inseparable samples.

As shown in Figure 1, for $(x_i, y_i)$ with $n$ samples, in which $x_i = [x_{i1}, x_{i2}, \cdots x_{in}]^T$, $y_i = [y_{i1}, y_{i2}, \cdots y_{in}]^T \in R^n$, SLFN with $L$ hidden layer nodes can be described as

$$\sum_{j=1}^{L} \eta_j \psi(\alpha_j \cdot x_i + \beta_j) = o_i, i = 1, 2, \cdots, n, \tag{1}$$

where $\alpha_j = [\alpha_{j1}, \alpha_{j2}, \cdots, \alpha_{jn}]^T$ and $\beta_j$ are input weight and threshold of hidden layer neuron. $\eta_j = [\eta_{j1}, \eta_{j2}, \cdots \eta_{jn}]^T$ is the connection weight between the $j$-th neuron in the hidden and the output layer, and $\psi(x)$ is the activation function of hidden layer neurons.

Formula (1) has the matrix form as:

$$\Psi \eta = Y, \tag{2}$$

where

$$\Psi(\alpha_1, \cdots, \alpha_L, \beta_1, \cdots, \beta_L, x_1, \cdots, x_N)$$
$$= \begin{bmatrix} \psi(\alpha_1 \cdot x_1 + \beta_1) & \psi(\alpha_2 \cdot x_1 + \beta_2) & \cdots & \psi(\alpha_L \cdot x_1 + \beta_L) \\ \psi(\alpha_1 \cdot x_2 + \beta_1) & \psi(\alpha_2 \cdot x_2 + \beta_2) & \cdots & \psi(\alpha_L \cdot x_2 + \beta_L) \\ \vdots & \vdots & & \vdots \\ \psi(\alpha_1 \cdot x_n + \beta_1) & \psi(\alpha_2 \cdot x_n + \beta_2) & \cdots & \psi(\alpha_L \cdot x_n + \beta_L) \end{bmatrix}_{n \times L} \tag{3}$$

and $\eta = [\eta_1, \eta_2, \cdots, \eta_L]_{L \times n}^T$, $Y = [y_1, y_2, \cdots, y_n]^T$.

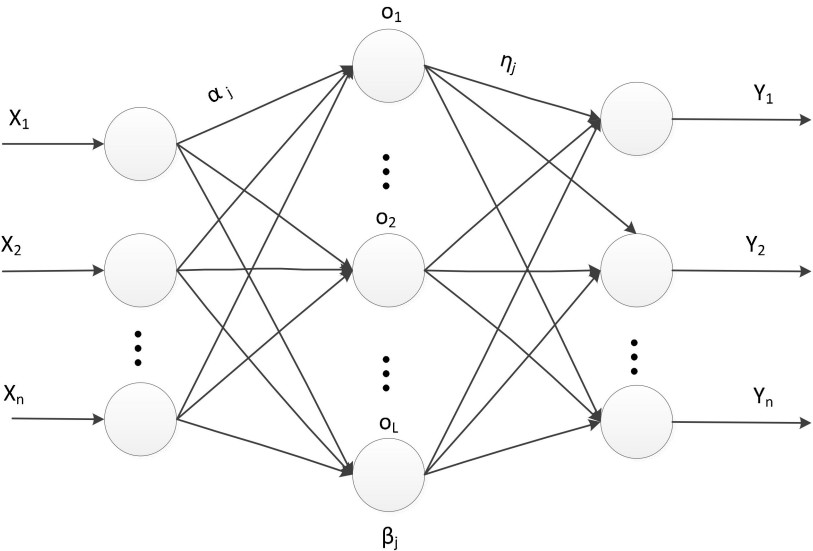

**Figure 1.** The basic structure of ELM.

Since the input weights are randomly generated, the hidden layer neuron threshold can also be randomly generated, and there is no need to adjust $\Psi$. The connection weight $\eta$ between the hidden and the output layer is obtained by solving the following formula:

$$\min_{\eta} \parallel \Psi\eta - Y \parallel, \tag{4}$$

Then:

$$\eta = \Psi^\dagger Y, \tag{5}$$

where $\Psi^\dagger$ is the Moore Penrose generalized inverse matrix of $\Psi$ [32,33].

KELM trains the network by the norm of the output weight and minimizes the training error. Then, from the standard optimization theory, the minimization obtained can be written as:

$$Min : L_p = \frac{1}{2} \parallel \eta \parallel^2 + C\frac{1}{2}\sum_{i=1}^{N} \vartheta_i^2 s.t. \psi(x_i)\eta = \mathbf{y}_i - \vartheta_i, i = 1, 2, \cdots, n, \tag{6}$$

where $C$ is the regularization parameter and $\vartheta_i$ is the training noise.

From the Karush Kuhn Tucker (KKT) theory [34], the above problem can be transformed into the optimization problem of formula (7):

$$L_{P_{kelm}} = \frac{\parallel \eta \parallel^2}{2} + \frac{C}{2}\sum_{i=1}^{N} \vartheta_i^2 - \sum_{i=1}^{N} \xi_i(\psi(x_i)\eta - \mathbf{y}_i + \vartheta_i), \tag{7}$$

where $\xi_i$ is the Lagrange operator.

By solving the optimization problem of above formula, we obtain:

$$\eta = \sum_{i=1}^{N} \xi_i \psi(x_i)^{\mathrm{T}} = \Psi^T \xi, \tag{8}$$

$$C\vartheta_i = \xi_i, i = 1, 2, \cdots, n, \tag{9}$$

$$\psi(x_i)\eta - \mathbf{y}_i + \vartheta_i = 0, i = 1, 2, \cdots, n, \tag{10}$$

where $\xi = [\xi_1, \xi_2, \cdots, \xi_n]^{\mathrm{T}}$.

Substituting (8) and (9) into (10), it holds that

$$\left(\Psi\Psi^T + \frac{1}{C}\right)\xi = Y. \tag{11}$$

$K(x_i, x_j) = \psi(x_i) \cdot \psi(x_j)^T$ $(i, j = 1, 2, \cdots, n)$ is the kernel function, the output of the KELM hidden layer can be written as:

$$f(x) = \Psi\eta = \Psi\Psi^T\left(\Psi\Psi^T + \frac{1}{C}\right)^{-1}Y. \tag{12}$$

According to the Mercer condition [35], the above formula can be written as:

$$f(x) = \begin{bmatrix} K(x, x_1) \\ K(x, x_2) \\ \vdots \\ K(x, x_N) \end{bmatrix}\left(\frac{1}{C} + K(x_i, x_j)\right)^{-1}Y, \tag{13}$$

where, the following Gaussian RBF kernel function [29,30] is used:

$$k(x_i, x_j) = \exp\left(-\frac{1}{2}\|x_i - x_j\|^2/\gamma^2\right) \tag{14}$$

with the parameter of the Gaussian RBF kernel function $\gamma$.

## 3. Indoor Positioning Algorithm Based on Reconstructed Observation Model and PF

### 3.1. Principle of Indoor Positioning Algorithm Based on Fingerprint Location

In indoor positioning scenarios, the signal propagation path of each AP is very complicated, which usually leads to low positioning accuracy. The indoor location method based on fingerprint positioning is relying on the smart mobile terminal to collect the RSS value of the reference nodes from the deployed APs. The RSS of each calibration node is significantly different. It can be uniquely identified by recording a set of RSS values of the calibration nodes. Then, the fingerprint information data of each reference node in the target area are collected in advance: thus, the reference node RSSI fingerprint database is established. Combined with the KELM algorithm, the nonlinear mapping relationship between the RSS signal and the corresponding position coordinates is established in order to achieve the purpose of positioning.

The indoor location algorithm based on location fingerprint mainly includes two stages: an offline stage and an online stage. At each reference node, the wireless signal strength of each AP is collected. Then the RSSI fingerprint data on the reference nodes are expressed as $(Z_i, R_i)(i = 1, 2, \cdots, N)$, where $Z_i = (x_i, y_i)$ is the spatial two-dimensional coordinates of the unknown node at the $i$-th moment, and $R_i = (RSSI_{i1}, RSSI_{i2}, \cdots, RSSI_{im})(i = 1, 2, \cdots, N)$ is the RSSI values composed of $m$ APs and is the signal strength vector tested at the $i$-th moment. Then the fingerprint database is established. Through filtering and training, the one-to-one correspondence between the RSSI high-dimensional vector and geographic location coordinate two-dimensional vector is obtained. In the online stage, a set of RSSI received is used to determine the final positioning coordinates through the trained model. The positioning principle based on RSSI data is shown in Figure 2 [36,37].

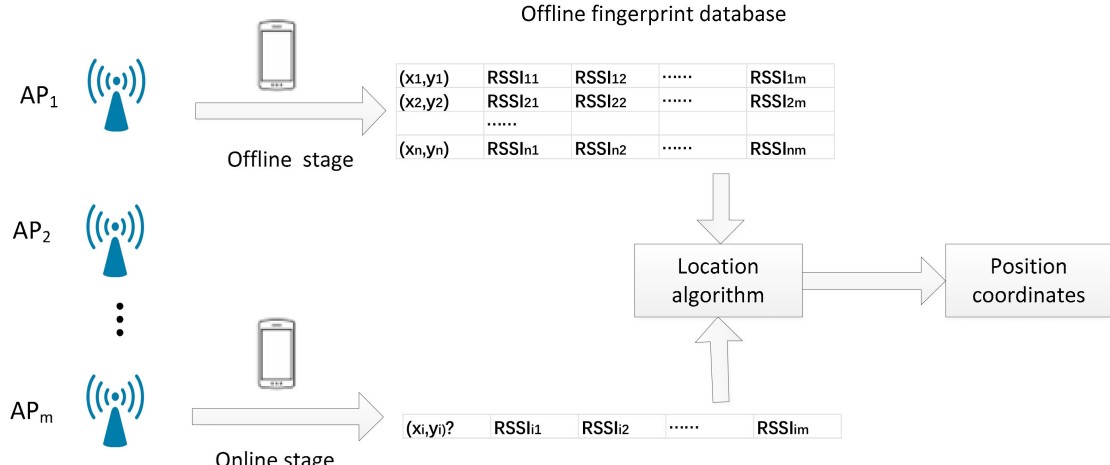

**Figure 2.** Location principle of fingerprint positioning. (The offline fingerprint database is established by collecting the wireless signal strength of the reference points in the offline stage. The positioning algorithm is the research method in this paper, and the position coordinates are the coordinates of unknown nodes).

Assuming that $n$ is the number of data samples collected and $m$ is the number of APs with unknown node receiving signal strength, the input matrix of the KELM can be described as:

$$
R = \begin{bmatrix} R_1 \\ R_2 \\ \vdots \\ R_n \end{bmatrix} = \begin{bmatrix} RSSI_{11} & RSSI_{12} & \cdots & RSSI_{1m} \\ RSSI_{21} & RSSI_{22} & \cdots & RSSI_{2m} \\ \vdots & \vdots & \ddots & \vdots \\ RSSI_{n1} & RSSI_{n2} & \cdots & RSSI_{nm} \end{bmatrix}_{n \times m}. \tag{15}
$$

The input weights with $L$-th hidden nodes are:

$$
\alpha = \begin{bmatrix} \alpha_1 \\ \alpha_2 \\ \vdots \\ \alpha_L \end{bmatrix} = \begin{bmatrix} \alpha_{11} & \alpha_{12} & \cdots & \alpha_{1n} \\ \alpha_{21} & \alpha_{22} & \cdots & \alpha_{2n} \\ \vdots & \vdots & \ddots & \vdots \\ \alpha_{L1} & \alpha_{L2} & \cdots & \alpha_{Ln} \end{bmatrix}_{L \times n}. \tag{16}
$$

The output of the networks can be expressed as:

$$
Z_i(x_i, y_i) = Y = \begin{bmatrix} x_1 & y_1 \\ x_2 & y_2 \\ \vdots & \vdots \\ x_n & y_n \end{bmatrix}. \tag{17}
$$

Therefore, the KELM model constructed is as follows:

$$
Z_i(x_i, y_i) = G(R_i), \tag{18}
$$

where $G(\cdot)$ is the mapping relationship between the unknown nodes position coordinates obtained by KELM training and the signal strength of each AP point received by the unknown nodes.

### 3.2. Particle Filter Localization and Receiving Factor Control Strategy

The determination of the location of unknown nodes indoors can be regarded as a random process of probability. Therefore, a PF algorithm [38] is used for positioning. In the indoor environment, $m$ APs with known locations need to be distributed. It is necessary

to establish a dynamic space model before using particle filtering to solve unknown node location. The position state of the unknown nodes is approximately modeled as:

$$X_i = \begin{bmatrix} 1 & 0 \\ 0 & 1 \end{bmatrix} X_{i-1} + u_{i-1} = FX_{i-1} + u_{i-1}, \tag{19}$$

where $X_i = [x_i, y_i]^{\mathrm{T}}$ is the two-dimensional coordinate of the unknown nodes and $F = \begin{bmatrix} 1 & 0 \\ 0 & 1 \end{bmatrix}$ is the discrete state transition matrix, and $u_{i-1}$ is the process noise obeying Gaussian distribution.

The observation model is constructed by KELM form formula (18):

$$\Gamma_i = HZ_i(x_i, y_i) + v_i = \begin{bmatrix} 1 & 0 \\ 0 & 1 \end{bmatrix} Z_i(x_i, y_i) + v_i, \tag{20}$$

where $H = \begin{bmatrix} 1 & 0 \\ 0 & 1 \end{bmatrix}$ represents the output matrix, and $v_i$ is the observation noise.

The procedure of the PF-based indoor positioning is shown in Algorithm 1, with $N_p$ as the particle number and $N_{vp}$ as the valid particles number.

The above-mentioned on indoor positioning methods based on KELM-PF assumes that the unknown nodes of the positioning system can normally and completely receive the signal strength of all APs. However, in complex indoor situations, the signal transmission will be blocked by walls, tables, chairs, partitions and other obstacles, and the unknown nodes cannot receive the signal strength of all nodes at the same time, so it is inevitable to lose part of the data of the APs. Therefore, how to design the location system has greatly practical value in the case of random data loss in wireless positioning system. At present, there are few research results on this problem. Reference [39] designed a robust filter for the FM model of a two-dimensional system when there exists data loss. However, for filtering under the circumstance that data loss based on the KELM-PF indoor positioning algorithm, the problem has not been studied yet. This paper proposes a control strategy by introducing a reception factor $E$ for the situation that unknown nodes in the wireless positioning system cannot receive the data of all APs, i.e., data loss occurs.

---

**Algorithm 1. PF-Based Indoor Positioning.**

---

Prediction: $X_i = FX_{i-1}$;
Predicted measurement: $\Gamma_i = HZ_i(x_i, y_i)$;
Input: set a threshold $N_{th}$;
for particle $k = 1 : N_p$ do
    Gaussian sampling: $\left\{ X_i^k \right\}_{k=1}^{N_p} = p\left( X_i \mid X_{i-1}^k \right)$;
    Calculate the weight for each particle $w_i^k = w_{i-1}^k \dfrac{p\left(\Gamma_i|X_i^k\right)p\left(X_i^k|X_{i-1}^k\right)}{q\left(X_i^k|X_{i-1}^k,\Gamma_{1:i}\right)}$;
end for
    Normalizing: $\hat{w}_i^k = w_i^k / \sum\limits_{k=1}^{N_p} w_i^k$;
    Important sampling: $N_{vp} = \left( \sum\limits_{k=1}^{N_p} \left(\omega_i^k\right)^2 \right)^{-1}$;
    If $N_{vp} > N_{th}$, important sampling;
    end if
    State estimation: $\hat{X}_i(x_i, y_i) \approx \sum\limits_{k=1}^{N_p} \hat{w}_i^k X_i$.

---

In the indoor positioning system mentioned above, $m$ APs are set. At $i$-th time, the signal strength of the unknown node that can receive the $m$ APs deployed, which is denoted as $R_i = (RSSI_{i1}, RSSI_{i2}, \cdots, RSSI_{im})$, $(i = 1, 2, \cdots, N)$. The number of AP data received

by the unknown node is denoted by $n_r$. Then a reception factor $E$ is introduced based on $n_r$: if $n_r < 3$, then set $E = 0$, and the indoor positioning algorithm uses the data of all APs received at the previous moment for PF calculating; if $n_r \geq 3$, then set $E = 1$. Then the KELM-PF algorithm is used for positioning. The expression of the reception factor $E$ is as follows:

$$E = \begin{cases} 1, n_r \geq 3 \\ 0, n_r < 3 \end{cases}. \tag{21}$$

### 3.3. Steps of Iterative Indoor Location Based on KELM-PF

This KELM-PF-based indoor positioning algorithm firstly trains the weights of the hidden and the output layer through the KELM network, then calculates the number $n_r$ of AP signal strengths received by unknown nodes, and adopts the control strategy of the reception factor $E$: if $E = 1$, using the KELM-PF algorithm reconstruct the observation model and then obtain the position coordinates of the unknown node; if $E = 0$, it returns to the last execution of PF.

The indoor positioning algorithm based on KELM-PF iteratively obtains its estimated value through the following Algorithm 2.

---

**Algorithm 2. KELM-PF Based Indoor Positioning**

---

**Inputs:** $\omega_j = [\omega_{j1}, \omega_{j2}, \cdots, \omega_{jn}]^T$, $\beta_j = [\beta_{j1}, \beta_{j2}, \cdots \beta_{jn}]^T$, $R_i$, $n_r$, $m$, $N_p$, $N_{th}$, $N_{vp}$;
**Step 1:** Training the weight parameters $\omega_j$ and $\beta_j$ of the hidden layer and output layer of KELM;
**Step 2:** Use the receiving factors $E$ to make decisions: calculate $n_r$. If $E = 1$, it will perform next step of the KELM-PF algorithm; if $E = 0$, jump to step 4 and execute the PF of the previous step;
**Step 3:** Execute the KELM-PF algorithm, reconstruct the observation model, obtain the observation value, and output the estimated value: put $R_i$ into the KELM network for training and obtain the observations. If $n_r = m$, put the signal strength vector $\widehat{R_i}$ obtained by real time detection into the KELM network for testing, and obtain the observed value. Then execute the particle algorithm of Algorithm 1 and output $\hat{X}_i(x_i, y_i)$; if $n_r < m$, set the signal strength of the unreceived node to 1, and then perform KELM testing. Then, execute the particle algorithm of Algorithm 1 and output $\hat{X}_i(x_i, y_i)$.
**Step 4:** Executes the PF of the previous step.

---

## 4. Experimental Results and Analysis

### 4.1. Verification of Validity

The experimental site is the laboratory on the fifth floor of the university: laboratory rooms 521, 520 and the corridor between them. The area of room 521 is about 80 m$^2$, the area of room 520 is about 60 m$^2$, and the area of the corridor is about 20 m$^2$, as shown in Figure 3. The wireless router is TP-LINK TL-R860, and eight wireless routers are placed in room 521 and room 520 (AP1-AP8 in Figure 3). The acquisition tool is the Wi-Fi fingerprint acquisition app, and all experiments are executed in the Matlab 2020a. We set up 66 Reference Nodes (RNs), including 30 RNs in room 521, 16 RNs in room 520 and 20 RNs in corridor. The distance between each RN is about 1.0 m. The black dots in Figure 3 represent the RNs used to collect RSS values in the offline stage. In each RN, we collect 60 samples with a time interval of 2 s, and a total of 18,000 sample data are obtained. Moreover, we set the number of training samples to 15,000 and the verification sample to 3000. $L$ is set to 100, $C$ is selected to 10, and 8 nodes are randomly selected to be tested in the rooms of 521, 520, and the corridor. Then the positioning accuracy obtained is shown in Figures 4–6. The Root Mean Square Error (RMSE) between the estimated position of the unknown nodes and the true position is calculated, and RMSE is the evaluation standard of the algorithm's performance:

$$RMSE = \sqrt{\frac{1}{M} \sum_{i=1}^{M} (x_i - x_{io})^2 + (y_i - y_{io})^2}, \tag{22}$$

where M is the Testing Node (TN) number, $(x_i, y_i)$ and $(x_{io}, y_{io})$ represent the predicted position coordinates and actual position coordinates of the unknown node, respectively.

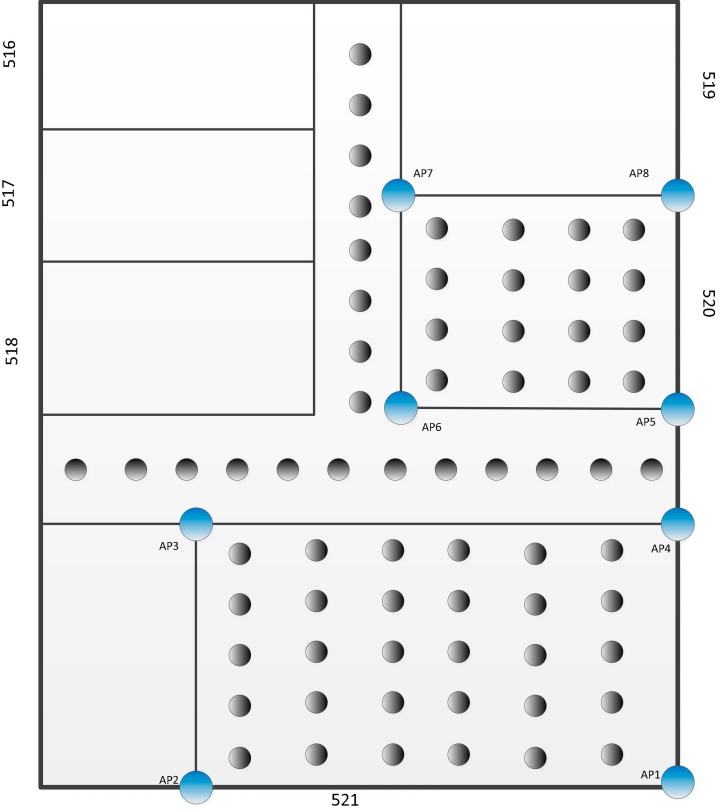

**Figure 3.** Nodes distribution map of indoor localization experiment area. ( AP1~AP8 are the 8 APs deployed; is the reference point for establishing the fingerprint database; 516~521 are the laboratory room numbers for indoor positioning).

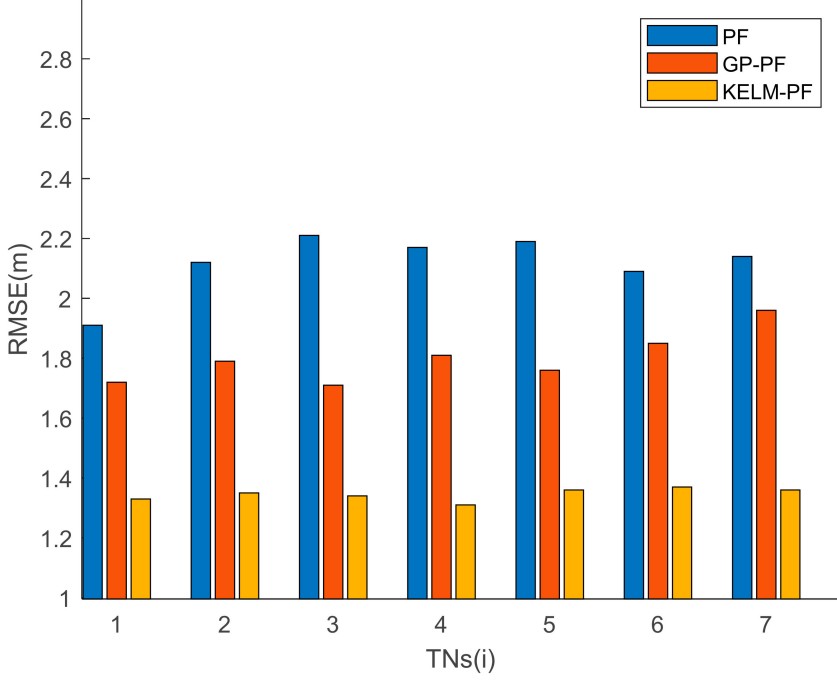

**Figure 4.** Positioning accuracies of different algorithms in room 521.

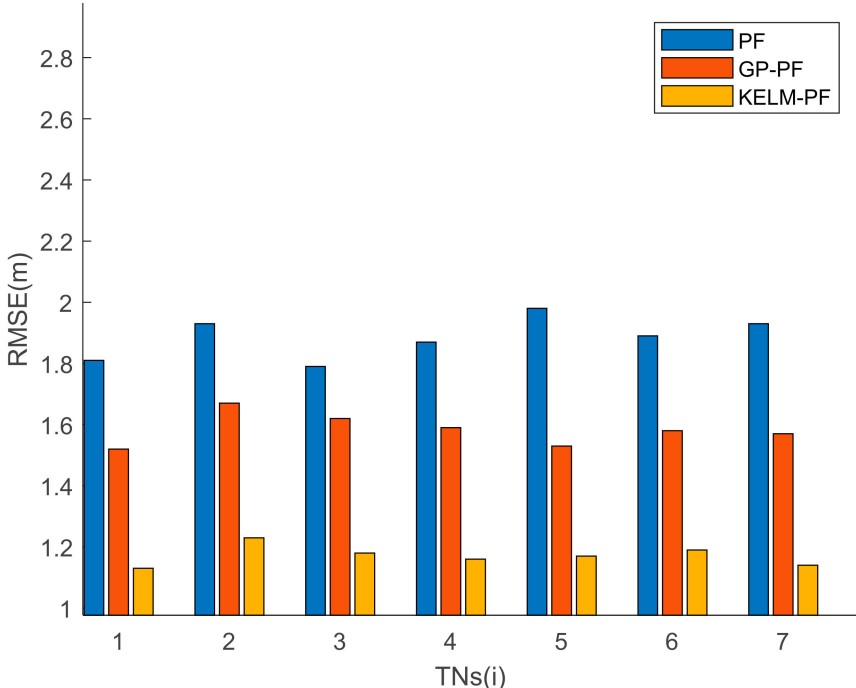

**Figure 5.** Positioning accuracies of different algorithms in room 520.

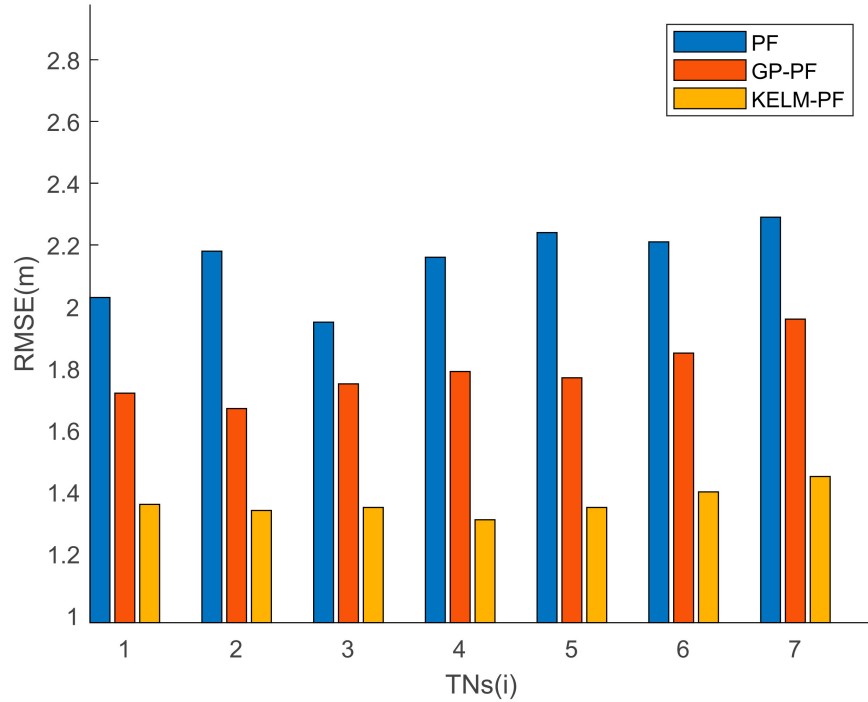

**Figure 6.** Positioning accuracies of different algorithms in the corridor.

The specific positioning accuracy comparison results of different algorithms in different indoor spaces are shown in Figures 4–6. The positioning accuracies of the PF, GP-PF and KELM-PF algorithms in room 521 are shown in Figure 4. The positioning accuracies of the PF, GP-PF and KELM-PF algorithms in room 520 are shown in Figure 5, and the positioning accuracies of the PF, GP-PF and KELM-PF algorithms in the corridor are shown in Figure 6. It can be seen from Figures 4–6 that the positioning accuracy of the KELM-PF algorithm is better than that of the PF and GP-PF algorithms. The KELM-PF algorithm has been tested in many places, and its positioning accuracy fluctuates little. Compared with a

single PF algorithm, the positioning accuracies in room 521, room 520 and the corridor are improved by 17.5%, 17.3%, and 16.1%, respectively.

### 4.2. Reference Node Density and Positioning Accuracy Experiments

As shown in Figure 3, four APs are placed in the laboratory 521. In the same environment, the same TNs are compared with its positioning accuracy at three densities of 0.5 m, 1.0 m, and 2.0 m. The KELM-PF algorithm estimated the position coordinates of TNs, and formula (22) is used to calculate the positioning errors at the three densities, as shown in Table 1. It can be seen from Table 1 that the positioning accuracies of the PF, GP-PF, and KELM-PF algorithms are related to the density of the fingerprint collection data. The smaller the distance between the collected data and the denser the fingerprint map database established, the higher the positioning accuracies. From the analysis of the results in Table 1, the positioning accuracies of KELM-PF is higher than that of GP-PF and PF, and the accuracies of GP-PF is slightly higher than that of PF. Meanwhile, we calculate standard deviation at an interval of 1.0 m with 400 samples. The standard deviations of the PF, GP-PF and KELM-PF are 0.072 m, 0.051 m and 0.032 m, respectively. The standard deviation of the KELM-PF algorithm is smallest, the one of the GP-PF algorithm is middle, and the one of the PF algorithm is the largest. It can be seen that the accuracies of positioning using the KELM-PF algorithm is higher than that of the GP-PF algorithm and the PF algorithm.

**Table 1.** Positioning accuracies of the same test nodes under different density.

|  | 0.5 m Intervals | 1.0 m Intervals | 2.0 m Intervals |
| --- | --- | --- | --- |
| PF | 0.74 | 1.36 | 2.25 |
| GP-PF | 0.65 | 1.12 | 2.08 |
| KELM-PF | 0.52 | 1.06 | 1.87 |

### 4.3. Comparison of Positioning Errors When PF Adopts Different Observation Models

The indoor positioning accuracies of the PF algorithm based on the logarithmic shadow propagation model, GP-PF and KELM-PF based observation model are compared. As shown in Figures 7 and 8, the PF positioning algorithm based on the KELM reconstruction observation model can track and locate faster than the GP-PF algorithm and the PF with the propagation model and has less fluctuations. Moreover, the positioning error of KELM-PF is smaller than that of PF and GP-PF, the tracking and positioning effect of KELM-PF is better than PF and GP-PF.

### 4.4. Analysis of Computational Complexity of Different Algorithms

The computational complexity of different algorithms is analyzed through running time of the positioning algorithm program on MATLAB (2020a). The computer is a ThinkPad notebook, the processor is an Intel(R) CORE(TM) i5-2520M CPU @ 2.1 GHz 2.5 GHz, the memory is 8 GB, and the operating system is Microsoft Windows 10 Professional Edition (64 bit). Table 2 lists the operating time loss of PF, GP-PF, and KELM-PF at 1.0 m and 2.0 m reference node densities. In the experiment, the number of nodes in the hidden layer of KELM is set to 100, and the number of particles of PF, GP-PF and KELM-PF is set to 300. The number of samples and the number of tests in the two experiments are the same. PF does not require offline training time, and positioning can be performed only in the online phase. It can be seen from Table 2 that GP-PF takes the longest time, and the KELM-PF algorithm is much faster than GP-PF in the offline training and online positioning phases and is slower than the PF algorithm.

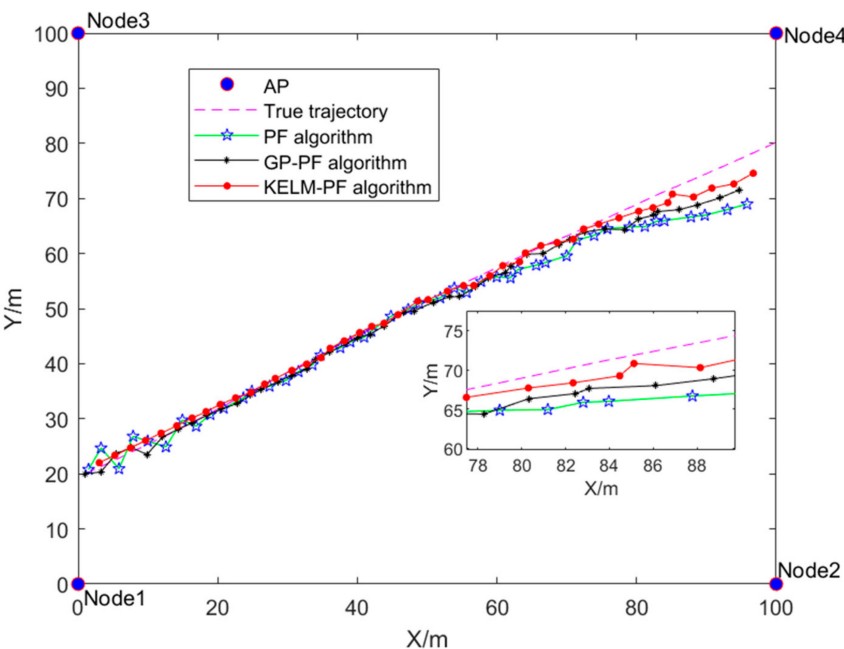

**Figure 7.** The position results of PF, GP-PF and KELM-PF.

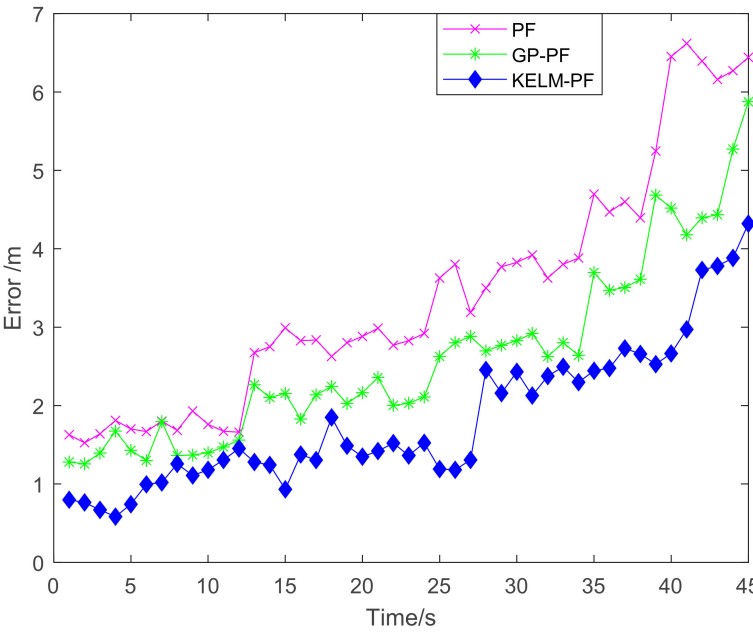

**Figure 8.** The RMSEs of PF, GP-PF and KELM-PF.

**Table 2.** Comparison of running time of different algorithms.

| Algorithm Name | Training Time(s) | | Testing Time(s) | |
|---|---|---|---|---|
| | 1.0 m Intervals | 2.0 m Intervals | 1.0 m Intervals | 2.0 m Intervals |
| PF | —— | —— | 0.096 | 0.083 |
| GP-PF | 0.482 | 0.576 | 0.531 | 0.396 |
| KELM-PF | 0.389 | 0.365 | 0.189 | 0.136 |

## 5. Conclusions

This paper proposes the indoor positioning algorithm based on reconstructed observation model by KELM training and PF. It firstly uses the SLFN KELM algorithm to establish nonlinear mapping relationship between unknown node positions and radio frequency

signal strength, optimizes it by the PF algorithm to realize indoor positioning, and proposes a control strategy based on the reception factor.

From the aspects of positioning accuracy, the relationship with fingerprint density, algorithm execution time, etc., we have conducted experiments, analysis and comparison. KELM-PF has higher positioning accuracy than GP-PF and PF. Furthermore, it is also verified that the indoor positioning accuracy of the KELM-PF algorithm is different under different reference node densities. The indoor positioning accuracy increases and improves in accordance with the reference node density. By analyzing the standard deviation of 1.0 m interval, it can be seen that the accuracy of positioning using the KELM-PF algorithm is higher than that of the GP-PF algorithm and the PF algorithm. However, the proposed KELM-PF indoor positioning still has shortcomings and needs further research. This method is only suitable for single-point positioning with little change in indoor positioning scenes. Therefore, in order to enhance the real-time and robustness of the KELM-PF algorithm, in the future we will study and discuss indoor positioning algorithms for multi-target dynamic environments in the future.

**Author Contributions:** This paper is the careful and collaborative work of all the authors. Li Ma wrote the paper and performed the experiments. Xiaoliang Feng suggested the main idea and the article structure. Ning Cao provided overall planning and important advice and supported the cognitive experiment. Jianping Zhang optimized the program code. Jingjing Yan performed experiments together with Li Ma and checked and modified the grammar and tense of the article. All authors have read and agreed to the published version of the manuscript.

**Funding:** This work was supported by the National Natural Science Foundation of China under Grant 61773154 and Grant U1804163.

**Institutional Review Board Statement:** Not applicable.

**Informed Consent Statement:** Not applicable.

**Data Availability Statement:** Part or all of the data, models or codes of the results of this research can be obtained from the corresponding author under reasonable request.

**Conflicts of Interest:** The authors declare no conflict of interest.

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
