# Peer review of "Indoor Positioning Algorithm Based on Reconstructed Observation Model and Particle Filter"

_ijgi, doi:10.3390/ijgi11010071_

Round 1
Reviewer 1 Report
The paper requires complete writing editing to make a user understand the technical details. As it is the paper lacks the presentation of a good research article, with sections beginning with examples rather than definitions and context. More background is needed to define and explain the technical terms used in the work. The benefits of the new system over other systems should be more clearly explained in the text. The variuos graphs should be explained clearly. There should be a DISCUSSIONS section before Conclusion to discuss the Results more in-depth and the significance of the work. The Conclusion should have more insights into the work done and limitations. It would be good to have a Diagrammatic representation of the methodology followed through a flowchart. After these modifications to the paper, it might be taken for possible publication.
Author Response
Dear Reviewer :
Thank you for your careful work and suggestions. I reply to your reviewer's comments as follows:
Point 1: The paper requires complete writing editing to make a user understand the technical details. As it is the paper lacks the presentation of a good research article, with sections beginning with examples rather than definitions and context. More background is needed to define and explain the technical terms used in the work.
Response 1: Thank you for your careful work and suggestions. We were very careful to revise the preface section and described the background of the research and the technical terms used in the work, see Introduction for details.
Point 2: The benefits of the new system over other systems should be more clearly explained in the text.
Response 2: Thank you for your careful work and suggestions. we described in detail the innovative points of the new method proposed In the Introduction, and further explained in the conclusion, see Introduction and Conclusion for details.
Point 3: The variuos graphs should be explained clearly.
Response 3: Thank you for your careful work and suggestions. We have carefully revised Fig.4, 5, 6, 7 and 8, as shown in Fig. 4-8.
Point 4: There should be a DISCUSSIONS section before Conclusion to discuss the Results more in-depth and the significance of the work.
Response 4: Thank you for your careful work and suggestions. In each part of the experiment, we analyzed and discussed the experimental results, see 4.1, 4.2, 4.3, 4.4. for details.
Point 5: The Conclusion should have more insights into the work done and limitations.
Response 5: Thank you for your careful work and suggestions. We summarized the work done in the conclusion and explained the limitations of the proposed method, see Conclusion for details.
Point 6: It would be good to have a Diagrammatic representation of the methodology followed through a flowchart.
Response 6: Thank you for your careful work and suggestions. We described the process through the steps of Algorithm 2 in section 3.3, see Algorithm 2 in section 3.3 for details.

Reviewer 2 Report
The paper proposes a Particle Filter (PF) indoor position algorithm based on the Kernel Extreme Learning Machine (KELM) reconstruction observation model. The paper must be improved both in terms of methodology description and technical presentation before it could be considered for publication.
Comments:
- The novelty of the paper is minor: there are several other works, which use WiFi RSS and particle filtering. The authors must clearly state their contribution and how it is different from previous works with appropriate references.
- An overview of related works is rather not structured. I rather suggest some general introduction to the problem of indoor navigation, discussion of the main approaches to the problem: infrastructure-based and infrastructure-less. Analyze and combine sources to identify trends, or gaps in the research. Then gradually go to the scope of approaches related to the approach proposed by the authors. In overall, the overview of the state-of-the-art is weak. Some of the discussed works are more than 20 years old and do not reflect current achievements in this area of research. More recent works on the use of BLEs for indoor localization should be discussed. I suggest “Dynamic indoor localization using maximum likelihood particle filtering”,” Fuzzy logic type-2 based wireless indoor localization system for navigation of visually impaired people in buildings”, and “Robust WiFi Localization by Fusing Derivative Fingerprints of RSS and Multiple Classifiers” to be discussed among others. Analyze the limitations of previous work and use it as a motivation for your own research.
- Line 104: “ELM is a new learning machine for SLFN …” -> It was proposed 15 years ago.
- ELM is a well-known method. Therefore, its description could be shortened and replaced by appropriate references. Instead, focus on the description of your own innovation.
- Subsection 3.2 just repeats some formulas from ref. [27]. Again, this is not necessary. The interested reader can find these formulas in [27]. I suggest to shorten and replace, perhaps, by an algorithm written in pseudocode.
- Present a distribution of spatial positioning errors as a histogram.
- It would be nice to evaluate the statistical reliability of the results. For example, you can present 95% confidence limits or standard deviation of numerical values given in Table 1 and Table 2.
- The comparison of the results with related work is missing. Provide (as a table) a comparison of accuracy achieved by your method with the results of other methods proposed by other authors.
- Discuss the limitations of the proposed method and threats-to-validity of the experimental results.
- Revise the conclusions. Be more explicit. Enlist the specific advantages of your method over similar methods and support your claims with main experimental results. Summarize the main advantages and limitations of the proposed method.
Author Response
Dear Reviewer:
Thank you for your careful work and suggestions. I reply to your reviewer's comments as follows:
Point 1: The novelty of the paper is minor: there are several other works, which use WiFi RSS and particle filtering. The authors must clearly state their contribution and how it is different from previous works with appropriate references.
Response 1: Thank you for your careful work and suggestions. This paper proposes a particle filter (PF) indoor positioning algorithm based on the kernel extreme learning machine (KELM) reconstructed observation model. There are two main aspects:
1) the use of KELM to construct the difference between the unknown node and the RF signal strength Linear mapping relationship, so as to reconstruct the observation model of indoor positioning, and combining with the PF algorithm to obtain the spatial two-dimensional coordinates of the unknown node;
2) By calculating the number of AP with signal strength received by unknown nodes, and introducing a control strategy of receiving factor E, it is possible to better solve the problem of data loss in the actual environment that unknown nodes cannot receive all AP data due to wireless signal congestion.
See Introduction for details.
Point 2: An overview of related works is rather not structured. I rather suggest some general introduction to the problem of indoor navigation, discussion of the main approaches to the problem: infrastructure-based and infrastructure-less. Analyze and combine sources to identify trends, or gaps in the research. Then gradually go to the scope of approaches related to the approach proposed by the authors. In overall, the overview of the state-of-the-art is weak. Some of the discussed works are more than 20 years old and do not reflect current achievements in this area of research. More recent works on the use of BLEs for indoor localization should be discussed. I suggest “Dynamic indoor localization using maximum likelihood particle filtering”,” Fuzzy logic type-2 based wireless indoor localization system for navigation of visually impaired people in buildings”, and “Robust WiFi Localization by Fusing Derivative Fingerprints of RSS and Multiple Classifiers” to be discussed among others. Analyze the limitations of previous work and use it as a motivation for your own research.
Response 2: Thank you for your careful work and suggestions. We have greatly modified Introduction. We discuss the main methods to solve the indoor positioning problem: infrastructure-based and infrastructure-less, and introduce the latest research results in infrastructure-based and infrastructure-less respectively, and then lead to the KELM-PF method and control strategy proposed in this paper, and delete inappropriate references,see Introduction for details.
Point 3: Line 104: “ELM is a new learning machine for SLFN …” -> It was proposed 15 years ago.
Point 4: ELM is a well-known method. Therefore, its description could be shortened and replaced by appropriate references. Instead, focus on the description of your own innovation.
Response 3 and 4: Thank you for your careful work and suggestions. We have modified the description and instructions of ELM, as shown in the second part of Kernel Extreme Learning Machine for details
Point 5: Subsection 3.2 just repeats some formulas from ref. [27]. Again, this is not necessary. The interested reader can find these formulas in [27]. I suggest to shorten and replace, perhaps, by an algorithm written in pseudocode.
Response 5: Thank you for your careful work and suggestions. We have modified section 3.2 and described it with pseudo-code algorithm 1. The details are shown in 3.2.
Point 6: Present a distribution of spatial positioning errors as a histogram.
Response 6: Thank you for your careful work and suggestions. According to your suggestion, we used a histogram to describe the spatial positioning error, as shown in Fig.4, Fig.5, and Fig.6.
Point 7: It would be nice to evaluate the statistical reliability of the results. For example, you can present 95% confidence limits or standard deviation of numerical values given in Table 1 and Table 2.
Response 7: Thank you for your careful work and suggestions. We calculated the standard deviations of the three methods of PF, GP-PF and KELM-PF in the experiment with an interval of 1.0 m in Table 1, and compared and analyzed the results. The details are shown in 4.2.
Point 8: The comparison of the results with related work is missing. Provide (as a table) a comparison of accuracy achieved by your method with the results of other methods proposed by other authors.
Response 8: Thank you for your careful work and suggestions. We have completed the experimental analysis and discussion in each experimental part, and the details are shown in 4.1, 4.2, 4.3, and 4.4.
Point 9: Discuss the limitations of the proposed method and threats-to-validity of the experimental results.
Response 9: Thank you for your careful work and suggestions. In each part of the experiment, we analyzed and discussed the experimental results, such as 4.1, 4.2, 4.3, 4.4; and the limitations of the method and the prospect of the next research were explained in Conclusion.
Point 10: Revise the conclusions. Be more explicit. Enlist the specific advantages of your method over similar methods and support your claims with main experimental results. Summarize the main advantages and limitations of the proposed method.
Response 10: Thank you for your careful work and suggestions. In Conclusion, we analyzed the advantages of the KELM-PF algorithm proposed in this paper over the particle filtering in reference [38] and GP-PF in reference [24] in terms of indoor localization accuracy, relationship with fingerprint density, algorithm execution time, etc., as detailed in Conclusion, as well as the limitations of the method and the outlook for the next research.

Reviewer 3 Report
The paper proposed a RSS-based method for deriving indoor position using KELM-PF algorithm. In particular, PF is used to compensate for possible data loss.
Overall, the topic of the paper sounds interesting. However, the writing style and many grammatical errors in the paper make it very hard to read thereby dwnplaying the merit of the work.
Some of the major points needed to be revised are the following.
- The step in 3.3 need to be more concise.
- The explanation of ELM and KELM from the equation 15 to 19 is very confusing. Specifically,
the definition of variable m and n are not specified. It is not clear why the dimension of Y
and n are m*2 and L*2. Also, x_k and y_k in equation 19 is not defined.
- In the experiment, details of the number of training and test samples shall be clearly described.
- In Fig 4-6, it is not clear what does Tps in x-axis stand for?
Author Response
Dear Reviewer:
Thank you for your careful work and suggestions. I reply to your reviewer's comments as follows:
Dear Reviewer:
Thank you for your careful work and suggestions. I reply to your reviewer's comments as follows:
Point 1: The step in 3.3 need to be more concise.
Response 1: Thank you for your careful work and suggestions. We simplified the steps in section 3.3 and used the algorithm 2 list to represent it, see Algorithm 2 in section 3.3 for details.
Point 2: The explanation of ELM and KELM from the equation 15 to 19 is very confusing. Specifically, the definition of variable m and n are not specified. It is not clear why the dimension of Y and n are m*2 and L*2. Also, x_k and y_k in equation 19 is not defined.
Response 2: Thank you for your careful work and suggestions. We modified the interpretation of KELM from equations 15 to 19 again and also gave the definitions of variables m and n. I am sorry that the x_k and y_k in equation 19 are wrong. A correction has been made, and the details are shown in formula (18), and the details are shown in section 3.1.
Point 3: In the experiment, details of the number of training and test samples shall be clearly described.
Response 3: Thank you for your careful work and suggestions. We clarified the number of training and testing samples for the experiment, as shown in section 4.1.
Point 4: In Fig 4-6, it is not clear what does Tps in x-axis stand for?
Response 4: Thank you for your careful work and suggestions. We modified Fig. 4-6 and modified the representation of the x-axis, as shown in Fig. 4-6.

Reviewer 4 Report
Dear Authors,
the article entitled: Indoor Positioning Algorithm Based on Reconstructed Observation Model and Particle Filter presents a Particle Filter (PF) indoor position algorithm based on the Kernel Extreme Learning Machine (KELM) reconstruction observation model. It is very important because the indoor positioning has been extensively used and developed with the increasing demand for indoor position information in recent years.
All chapters (abstract, introduction, Kernel Extreme Learning Machine, indoor positioning algorithm based on reconstructed observation model and PF, experimental results and analysis, as well as conclusions) are very well described and they do not raise any doubts. In terms of the literature review is not complex (28 positions), all of which are research articles from recognized scientific journals, such as: International Journal of Distributed Sensor Networks, IEEE Transactions on Industrial Electronics, Sensors, and others. Moreover, I would like to point out that the publications cited are related to the subject of this article (indoor positioning, KELM, PF and reconstructed observation model). However, in the paper make the following changes:
- The abstract and conclusions should provide detailed results of experimental studies.
- I propose to extend the literature in the first paragraph of the introduction, related to the applications of GNSS in outdoor environments that are on the Journal Citations Reports (JCR) list such as for example:
- Figiel, S.; Specht, C.; Moszyński, M.; Stateczny, A.; Specht, M. Testing of Software for the Planning of a Linear Object GNSS Measurement Campaign under Simulated Conditions. Energies 2021, 14, 7896.
- Gao, Z.; Ge, M.; Li, Y.; Shen, W.; Zhang, H.; Schuh, H. Railway Irregularity Measuring Using Rauch–Tung–Striebel Smoothed Multi-sensors Fusion System: Quad-GNSS PPP, IMU, Odometer, and Track Gauge. GPS Solut. 2018, 22, 36.
- Krasuski, K.; Savchuk, S. Determination of the Precise Coordinates of the GPS Reference Station in of a GBAS System in the Air Transport. Commun. Sci. Lett. Univ. Zilina 2020, 22, 11–18.
- In the axes of Figures 4-6, SI units are missing.
- Please write sentences impersonally.
To sum up, after taking into account the above amendments (minor revision), I suppose that this article is suitable for publication in the ISPRS International Journal of Geo-Information.
Author Response
Dear Reviewer:
Thank you for your careful work and suggestions. I reply to your reviewer's comments as follows:
Point 1: The abstract and conclusions should provide detailed results of experimental studies.
Response 1: Thank you for your careful work and suggestions. We carefully revised the abstract and conclusions, and elaborated the experimental research results in more detail, see Abstract and Conclusions for details.
Point 2: I propose to extend the literature in the first paragraph of the introduction, related to the applications of GNSS in outdoor environments that are on the Journal Citations Reports (JCR) list such as for example:
- Figiel, S.; Specht, C.; Moszyński, M.; Stateczny, A.; Specht, M. Testing of Software for the Planning of a Linear Object GNSS Measurement Campaign under Simulated Conditions. Energies 2021, 14, 7896.
- Gao, Z.; Ge, M.; Li, Y.; Shen, W.; Zhang, H.; Schuh, H. Railway Irregularity Measuring Using Rauch–Tung–Striebel Smoothed Multi-sensors Fusion System: Quad-GNSS PPP, IMU, Odometer, and Track Gauge. GPS Solut. 2018, 22, 36.
- Krasuski, K.; Savchuk, S. Determination of the Precise Coordinates of the GPS Reference Station in of a GBAS System in the Air Transport. Commun. Sci. Lett. Univ. Zilina 2020, 22, 11–18.
Response 2: Thank you for your careful work and suggestions. We have expanded the literature related to the application of GNSS in outdoor environments in the preface, such as the first paragraph of Introduction.
Point 3: In the axes of Figures 4-6, SI units are missing.
Response 3: Thank you for your careful work and suggestions. We modified Fig. 4-6 and added the coordinate units, see Fig. 4-6 for details.
Point 4: Please write sentences impersonally.
Response 4: Thank you for your careful work and suggestions. We checked the grammar and polished the language of the full text.

Round 2
Reviewer 2 Report
The authors have addressed all my suggestions and concerns, and have revised the manuscript accordingly. The quality and soundness of the manuscript were improved. I believe the article would make a valuable contribution to the journal and recommend it to be accepted.
Reviewer 3 Report
Thank you for addressing all the comments.